# LayoutGAN: Generating Graphic Layouts with Wireframe Discriminators

**Jianan Li**[1]*,    **Jimei Yang**[2],    **Aaron Hertzmann**[2],    **Jianming Zhang**[2],    **Tingfa Xu**[1]

1. Beijing Institute of Technology        2. Adobe Research

{20090964,xutingfa}@bit.edu.cn, {jimyang,hertzman,jianmzha}@adobe.com

## Abstract

Layout is important for graphic design and scene generation. We propose a novel Generative Adversarial Network, called LayoutGAN, that synthesizes layouts by modeling geometric relations of different types of 2D elements. The generator of LayoutGAN takes as input a set of randomly-placed 2D graphic elements and uses self-attention modules to refine their labels and geometric parameters jointly to produce a realistic layout. Accurate alignment is critical for good layouts. We thus propose a novel differentiable wireframe rendering layer that maps the generated layout to a wireframe image, upon which a CNN-based discriminator is used to optimize the layouts in image space. We validate the effectiveness of Layout-GAN in various experiments including MNIST digit generation, document layout generation, clipart abstract scene generation and tangram graphic design.

## 1 Introduction

Graphic design is an important visual communication tool in our modern world, encompassing everything from book covers to magazine layouts to web design. Whereas methods for generating realistic natural-looking images have made significant progress lately, particularly with Generative Adversarial Networks (GANs) (Karras et al., 2018), methods for creating designs are far more primitive. This is, in part, due to the difficulty of finding data representations suitable for learning. Graphic designs are normally composed of vector representations of primitive objects, such as polygons, curves and ellipses, instead of pixels laid on a regular lattice. The quality and content of a design depends on the presence of elements, their attributes, and their relations to other elements. The visual perception of design depends on the arrangement of these elements; misalignment of two elements of just a few millimeters can ruin the design. Training from images of designs using conventional GANs synthesizes the layouts in pixel space, and thus mixes up layout and its rendering, and thus would be unlikely to capture layout styles well. Modeling such highly-structured data using neural networks is of great interest as they usually represent human abstract knowledge about the visual world (Zitnick & Parikh, 2013; Song et al., 2017) and how this knowledge are expressed via documents and designs (Deka et al., 2017; Yang et al., 2017).

This paper introduces LayoutGAN, a novel GAN which directly synthesizes a set of graphical elements in a design. In a given design problem, a fixed set of element classes (e.g., "title," "figure") is specified in advance. In our network, each element is represented by its class probabilities and its geometric parameters, i.e., bounding-box keypoints. The generator takes as input graphic elements with randomly-sampled class probabilities and geometric parameters, and arranges them in a design; the output is the refined class probabilities and geometric parameters of the design elements. The generator has the desirable property of being permutation-invariant: it will generate the same layout if we re-order the input elements.

We propose two kinds of discriminator networks for this structured data. The first is similar in structure to the generator: it operates directly on the class probabilities and geometric parameters of the elements. Though effective, it is not sensitive enough to misalignment and occlusion between elements. The second discriminator operates in the visual domain. Like a human viewer who judges

---

*Work done during Jianan Li's internship at Adobe Research.

a design by looking at its rasterized image, the relations among different elements can be evaluated well by mapping them to 2D layouts. Then Convolutional Neural Networks (CNNs) can be used for layout optimization as they are specialized in distinguishing visual patterns including but not limited to misalignment and occlusion. However, the key challenge is how to map the geometric parameters to pixel-level layouts differentiably. One approach would be to render the graphic elements into bitmap masks using Spatial Transformer Networks (Jaderberg et al., 2015). But we found that filled pixels within the design elements cause occlusions and are ineffective for back-propagation, for example, when a small polygon hides behind a larger one. We experimented with bitmap mask rendering but it was not successful. In this paper, we propose a novel differentiable wireframe rendering layer that rasterizes both synthesized and real structured data of graphic elements into wireframe images, upon which a standard CNN can be used to optimize the layout across the visual and the graphic domain. The wireframe rendering discriminator has several advantages. First, convolution layers are very good at extracting spatial patterns of images so that they are more sensitive to alignment. Second, the rendered wireframes make elements visible even when they overlap and thus the network is alleviated from inferring the occlusions that may occur in other renderings such as masks.

We evaluate the LayoutGAN for several different tasks, including a sanity test on MNIST digits, generating page layouts from labeled bounding boxes, generating clipart abstract scenes, tangram graphic design, and mobile app design layouts. In each case, our method successfully generates layouts respecting the types of elements and their relationships for the problem domain.

In summary, the LayoutGAN comprises the following contributions: 1. A Generator that directly synthesizes structured data, represented as a resolution-independent set of labeled graphic elements in a design. 2. A differentiable wireframe rendering layer which allows the Discriminator to judge alignment from discrete element arrangements.

## 2 RELATED WORK

**Structured data generation.** Convolutional networks have been shown successful for generating data in regular lattice, such as images (Radford et al., 2015), videos (Vondrick et al., 2016) and 3D volumes (Yan et al., 2016; Wu et al., 2016). When generating highly-structured data, such as text (Donahue et al., 2015) and programs (Reed & De Freitas, 2015), recurrent networks are often the first choice (Sutskever et al., 2014), especially equipped with attention (Bahdanau et al., 2014) and memory modules (Graves et al., 2014). Recently, researchers show that convolutional networks can be also used to synthesize sequences (Oord et al., 2016; van den Oord et al., 2016) using auto-regressive models. However, in many cases, an object has no sequential order (Vinyals et al., 2015), but is a set of elements, e.g. point clouds. Fan et al. (2017) propose a point set generation network for synthesizing 3D point clouds of the object shape from a single images. It is further paired with a point set classification network (Charles et al., 2017) for auto-encoding 3D point clouds (Achlioptas et al., 2017). Our work extends the set representation to more general primitive objects, i.e., labeled polygons. Meanwhile, researchers also model the structured data of connected elements using graph convolutions (Kipf & Welling, 2017).

**Data-driven graphic design.** Automating layout is a classic problem in graphic design (Hurst et al., 2009). O'Donovan et al. (2014) formulate an energy function by assembling various heuristic visual cues and design principles to optimize single-page layouts, and extend this to an interactive tool (O'Donovan et al., 2015). The model parameters are learned from a small number of example designs. Pang et al. (2016) optimize layout for desired gaze direction. Deka et al. (2017) collect a mobile app design database for harnessing data-driven applications and present preliminary results of learning similarities of pixel-level textural/non-textual masks for design search, but do not learn models from this data. Swearngin et al. (2018) propose an interactive system that converts example design screenshots to vector graphics for designers to re-use and edit. Bylinskii et al. (2017) analyze the visual importance of graphic designs and use saliency map as the driving force to assist retargeting and thumbnailing. Previous methods have learned models for other graphic design elements, such as fonts (O'Donovan et al., 2014) and colors (O'Donovan et al., 2011). These are orthogonal to the layout problem, and could be combined in future work. No previous method has learned to create design or layout from large datasets, and no previous work has applied GANs to layout.

**3D scene synthesis.** Interior scene synthesis and furniture layout generation draws great interest in graphics community. Early approaches focus on optimization of hand-crafted design princi-

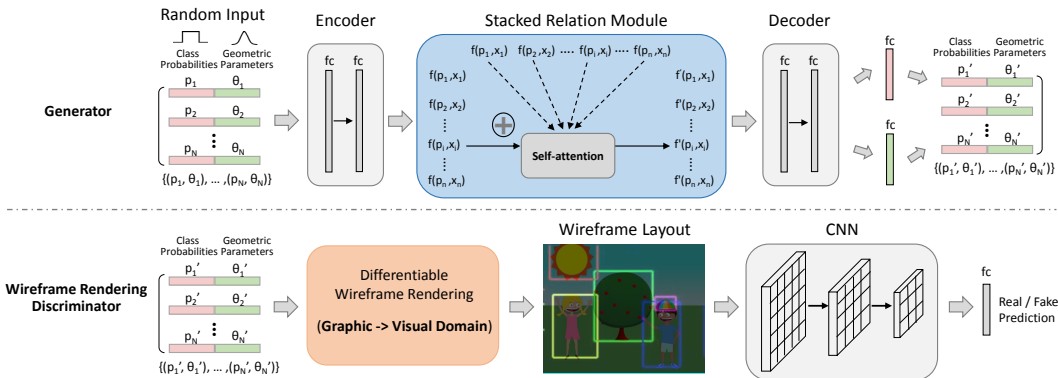

Figure 1: Overall architecture of LayoutGAN. The generator takes as input graphic elements with randomly sampled class probabilities and geometric parameters from Uniform and Gaussian distribution respectively. An encoder embeds the input and feeds them into the stacked relation module, which refines the embedded features of each element in a coordinative manner by considering its semantic and spatial relations with all the other elements. Finally, a decoder decodes the refined features back to class probabilities and geometric parameters. The wireframe rendering discriminator feeds the generated results to a differentiable wireframe rendering layer which raterizes the input graphic elements into 2D wireframe images, upon which a CNN is applied for layout optimization.

ples (Merrell et al., 2011) and learning statistical priors of pairwise object relationships (Fisher et al., 2012) due to limited data. Wang et al. (2018a) recently proposed a sequential decision making approach to indoor scene synthesis. In each step, a CNN is trained to predict either location or category of one object by looking at the rendered top-down views. This resembles our wireframe rendering discriminator, in the sense of using convolutions to capture spatial patterns of layouts.

## 3 LAYOUTGAN

This section describes our data and model representations.

### 3.1 DESIGN REPRESENTATION

In our model, a graphic design is comprised of a set of $N$ primitive design elements $\{(\boldsymbol{p_1}, \boldsymbol{\theta_1}), \cdots, (\boldsymbol{p_N}, \boldsymbol{\theta_N})\}$. Each element has a set of geometric parameters $\boldsymbol{\theta}$, and a vector of class probabilities $\boldsymbol{p}$. The entries of these variables are problem-dependent. For example, document layouts include 6 classes, e.g. "title" and "picture" while clipart layouts include 6 classes, e.g. "boy" and "hat". For 2D point set generation (in MNIST digits), $\boldsymbol{\theta} \equiv [x, y]$, representing the coordinates of each point; for bounding-box generation in document layout, $\boldsymbol{\theta} \equiv [x^L, y^T, x^R, y^B]$, representing the top-left and bottom-right coordinates of each bounding box; for layouts with scale and flip (clipart abstract scenes), $\boldsymbol{\theta} \equiv [x, y, s, l]$, representing the center coordinates, scale and flip of each element.

### 3.2 GENERATOR ARCHITECTURE

In the LayoutGAN, the Generator is a function $\boldsymbol{G(z)}$ that takes a layout as input, where $\boldsymbol{z} = \{(\boldsymbol{p_1}, \boldsymbol{\theta_1}), \cdots, (\boldsymbol{p_N}, \boldsymbol{\theta_N})\}$ consisting of initial graphic elements with randomly-sampled geometric parameters $\boldsymbol{\theta_i}$, and one-hot encoding of a randomly-sampled class $\boldsymbol{p_i}$. The Generator outputs a refined layout $\boldsymbol{G(z)} = \{(\boldsymbol{p'_1}, \boldsymbol{\theta'_1}), \cdots, (\boldsymbol{p'_N}, \boldsymbol{\theta'_N})\}$ which is meant to resemble a real graphic design. Note that, unlike, conventional GANs where $\boldsymbol{z}$ represents a low-dimensional latent variable, our $\boldsymbol{z}$ represents initial random graphic layout that has the same structure as the real one. The Discriminator learns to capture the geometric relations among different types of elements for layout optimization from both the graphic and the visual domain. Next, we go into details of the generator and discriminator design.

As illustrated in Figure 1, the generator takes as input a set of graphic elements with random class probabilities and geometric parameters sampled from Uniform and Gaussian distribution respectively. An encoder consisting of a multilayer perceptron network (implemented as multiple fully connected layers) first embeds the class one-hot vectors and geometric parameters of each graphic element. The relation module, implemented as self-attention inspired by Wang et al. (2018b), is then used to embed the feature of each graphic element as a function of its spatial context, i.e., its

relations with all the other elements in the design. Denote $f(\boldsymbol{p_i}, \boldsymbol{\theta_i})$ as the embedded feature of the graphic element $i$, its refined feature representation $f'(\boldsymbol{p_i}, \boldsymbol{\theta_i})$ can be obtained through a contextual residual learning process, which is defined as:

$$f'(\boldsymbol{p_i}, \boldsymbol{\theta_i}) = \boldsymbol{W_r} \frac{1}{N} \sum_{\forall j \neq i} \boldsymbol{H}(f(\boldsymbol{p_i}, \boldsymbol{\theta_i}), f(\boldsymbol{p_j}, \boldsymbol{\theta_j})) \boldsymbol{U}(f(\boldsymbol{p_j}, \boldsymbol{\theta_j})) + f(\boldsymbol{p_i}, \boldsymbol{\theta_i}), \tag{1}$$

where an unary function $\boldsymbol{U}$ computes a representation of the embedded feature $f(\boldsymbol{p_j}, \boldsymbol{\theta_j})$ of element $j$ and a pairwise function $\boldsymbol{H}$ computes a scalar value representing the relation between elements $i$ and $j$. Thus, all the other elements $j \neq i$ contribute to the feature refinement of element $i$ by summing up their relations. The response is normalized by the total number of elements in the set, $N$. The weight matrix $\boldsymbol{W_r}$ computes a linear embedding, producing the contextual residual to be added to $f(\boldsymbol{p_i}, \boldsymbol{\theta_i})$ for feature refinement. In our experiments, we define $\boldsymbol{H}$ as a dot-product:

$$\boldsymbol{H}(f(\boldsymbol{p_i}, \boldsymbol{\theta_i}), f(\boldsymbol{p_j}, \boldsymbol{\theta_j})) = \boldsymbol{\psi}(f(\boldsymbol{p_i}, \boldsymbol{\theta_i}))^T \boldsymbol{\phi}(f(\boldsymbol{p_j}, \boldsymbol{\theta_j})), \tag{2}$$

where $\boldsymbol{\psi}(f(\boldsymbol{p_i}, \boldsymbol{\theta_i})) = \boldsymbol{W_\psi} f(\boldsymbol{p_i}, \boldsymbol{\theta_i})$ and $\boldsymbol{\phi}(f(\boldsymbol{p_j}, \boldsymbol{\theta_j})) = \boldsymbol{W_\phi} f(\boldsymbol{p_j}, \boldsymbol{\theta_j})$ are two linear embeddings. We stack 4 relation modules for feature refinement in our experiments. Finally, a decoder consisting of another multilayer perceptron network followed by two branches of fully connected layer with a sigmoid activation function is used to map the refined feature of each element back to class probabilities and geometric parameters respectively. Optionally, Non-Maximum Suppression (NMS) can be applied to remove duplicated elements.

### 3.3 DISCRIMINATOR NETWORK ARCHITECTURES

The discriminator aims to distinguish between synthesized and real layouts. We present two types of discriminators, one based on relation modules directly built upon layout parameters, and the other based on how the layouts look like via rendering.

#### 3.3.1 RELATION-BASED DISCRIMINATOR

The Relation-Based Discriminator takes as input a set of graphic elements represented by class probabilities and geometric parameters, and feeds them to an encoder consisting of a multilayer perceptron network for feature embedding $f(\boldsymbol{p_i}, \boldsymbol{\theta_i})$. It then extracts their global graphical relations $\boldsymbol{g}(\boldsymbol{r}(\boldsymbol{p_1}, \boldsymbol{\theta_1}), \cdots, \boldsymbol{r}(\boldsymbol{p_N}, \boldsymbol{\theta_N}))$ where $\boldsymbol{r}(\boldsymbol{p_i}, \boldsymbol{\theta_i})$ represents a simplified relation module as in equation 1 by removing the shortcut connection, and $\boldsymbol{g}$ is a max-pooling function used in (Charles et al., 2017). Thus, the global relations among all the graphic elements can be modeled, upon which a classifier composed by a multilayer perception network is applied for real/fake prediction.

#### 3.3.2 WIREFRAME RENDERING DISCRIMINATOR

The Wireframe Rendering Discriminator exploits CNNs to classify layouts, in order to learn to classify the visual properties of a layout. The Discriminator consists of a wireframe rendering layer, which produces an output image $\boldsymbol{I}$, which is then fed into a CNN for classification.

The rasterization is performed as follows. Suppose there are $N$ elements in the design, with parameters $\{(\boldsymbol{p_1}, \boldsymbol{\theta_1}), ..., (\boldsymbol{p_N}, \boldsymbol{\theta_N})\}$. Each element can be rendered into its own grayscale image $\boldsymbol{F_\theta}(x, y)$; rendering details for specific types of elements are given below. The image dimensions for each individual $\boldsymbol{F}$ are $W \times H$, where $W$ and $H$ are the width and height of the design in pixels.

The layer output is a multi-channel image $\boldsymbol{I}$ of dimensions $W \times H \times M$, where each channel corresponds to one of the $M$ element types. In other words, pixel $(x, y)$ of $\boldsymbol{I}$ is a class activation vector for that pixel, and is computed as:

$$\boldsymbol{I}(x, y, c) = \max_{i \in [1...N]} p_{i,c} \boldsymbol{F_{\theta_i}}(x, y) \tag{3}$$

where $p_{i,c}$ is the probability that element $i$ is of class $c$.

We next describe the rendering process for computing $\boldsymbol{F_\theta}$, in the cases where $\boldsymbol{\theta}$ represents a point, a rectangle, and a triangle.

**Point.** We start with the simplest geometric form, a single keypoint $\boldsymbol{\theta_i} = (x_i, y_i)$ for element $i$. We implement an interpolation kernel $k$ for its rasterization. Its spatial rendering response on $(x, y)$ in the rendered image can be written as:

$$\boldsymbol{F_{\theta_i}}(x, y) = k(x - x_i) k(y - y_i). \tag{4}$$

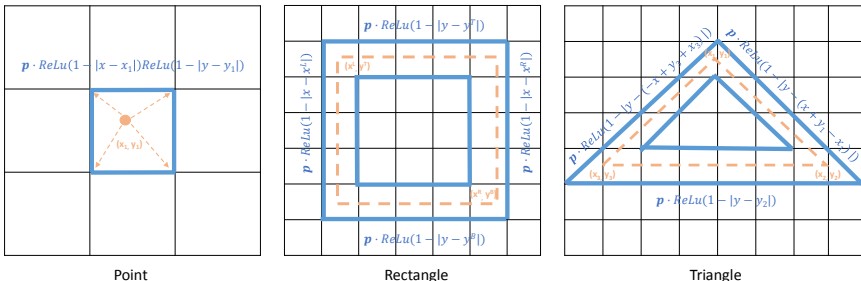

Figure 2: Wireframe rendering of different polygons (point, rectangle and triangle). The black grids represent grids of target image. The orange dots/dotted lines represent the graphic element mapped onto the image grid. The blue solid lines represent the rasterized wireframes expressed as differentiable functions of graphic elements in terms of both class probilities and geometric parameters.

We adopt bilinear interpretation (Johnson et al., 2016), corresponding to the kernel $k(d) = \max(0, 1 - |d|)$ (implemented as ReLU activation), as shown in Figure 2. As $I$ is a differentiable function of the class probabilities and the coordinates, the sub-gradients of the rasterized image can be thus propagated backward to them. We validate such rendering design for MNIST digit generation, detailed experiments can be seen in Section 4.1.

**Rectangle.** We now consider more complex polygons. Assuming an element is a rectangle, or bounding box represented by its top-left and bottom-right coordinates $\theta = (x^L, y^T, x^R, y^B)$, which is very common in various designs. Specifically, considering a rectangle $i$ with coordinates $\theta_i = (x_i^L, y_i^T, x_i^R, y_i^B)$, as shown in Figure 2, the black grids represent the locations in the rendered image and the orange dotted box represents the rectangle being rasterized in the rendered image. For a wireframe representation, only the points near the boundary of the dotted box (lie in blue solid line) are related to the rectangle, so its spatial rendering response on $(x, y)$ can be formulated as:

$$\boldsymbol{F}_{\boldsymbol{\theta_i}}(x, y) = \max \begin{pmatrix} k(x - x_i^L)b(y - y_i^T)b(y_i^B - y), \\ k(x - x_i^R)b(y - y_i^T)b(y_i^B - y), \\ k(y - y_i^T)b(x - x_i^L)b(x_i^R - x), \\ k(y - y_i^B)b(x - x_i^L)b(x_i^R - x) \end{pmatrix}, \tag{5}$$

where $b(d) = \min(\max(0, d), 1)$ constraining the rendering to nearby pixels.

**Triangle.** We further describe the wireframe rendering process of another geometric form, triangle. For triangle $i$ represented by its three vertices' coordinates $\theta_i = (x_i^1, y_i^1, x_i^2, y_i^2, x_i^3, y_i^3)$, when $x_i^1 \neq x_i^2 \neq x_i^3$, its spatial rendering response on $(x, y)$ in the rendered image can be calculated as:

$$\boldsymbol{F}_{\boldsymbol{\theta_i}}(x, y) = \max \begin{pmatrix} k(y - \frac{(y_i^2 - y_i^1) \cdot (x - x_i^1)}{x_i^2 - x_i^1} - y_i^1)b(x - x_i^1)b(x_i^2 - x), \\ k(y - \frac{(y_i^3 - y_i^1) \cdot (x - x_i^1)}{x_i^3 - x_i^1} - y_i^1)b(x - x_i^3)b(x_i^1 - x), \\ k(y - \frac{(y_i^3 - y_i^2) \cdot (x - x_i^2)}{x_i^3 - x_i^2} - y_i^2)b(x - x_i^3)b(x_i^2 - x) \end{pmatrix}, \tag{6}$$

Through this wireframe rendering process, gradients can be propagated backward to both the class probabilities and geometric parameters of the graphic elements for joint optimization.

A CNN consisting of 3 convolutional layers, followed by a fully connected layer with sigmoid activation, is applied to the rasterization layer $I$ to predict real/fake graphic layouts.

## 4 EXPERIMENTS

The implementation is based on TensorFlow (Abadi et al., 2016). The network parameters are initialized from zero-mean Gaussian with standard deviation of 0.02. All the networks are optimized using Adam (Kingma & Ba, 2014) with a fixed learning rate of 0.00002. Detailed architectures can be found in the appendix.

### 4.1 MNIST DIGIT GENERATION

As a toy problem, we generate MNIST layouts. MNIST is a handwritten digit database consisting of 60,000 training and 10,000 testing images. For each image, we extract the locations of 128 randomly-selected foreground pixels as the graphic representation so that digit generation can be

Relation-based    Wireframe Rendering    Rendered GT    Initial Location    Refined Location

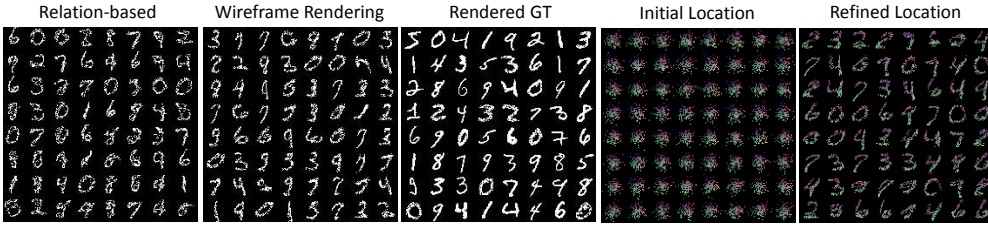

(a) Comparing generated MNIST samples.        (b) Point tracing (best viewed in color).

Figure 3: Results on MNIST digit generation.

Table 1: Inception scores for generated digits.

| Methods | Score $\pm$ std |
|---|---|
| Relation | $6.53 \pm .09$ |
| Wireframe | $7.36 \pm .07$ |
| Real data | $9.81 \pm .08$ |

Table 2: Spatial analysis of document layout.

| Methods | Overlap(%) | Alignment(%) |
|---|---|---|
| Relation | 1.52 | 6.4 |
| Wireframe | 1.17 | 3.4 |
| Real data | 0.05 | 0.5 |

formulated as generating of 2D point layouts. In Figure 3a, each image shows $8 \times 8$ digits rendered from point layouts. The left and middle images are the generated samples from the LayoutGAN with a relation-based discriminator and a wireframe rendering discriminator, respectively. The right image shows the digits rendered from ground-truth point layouts. One can see that the LayoutGAN with either discriminator captures various patterns. The wireframe rendering discriminator generates more compact, better-aligned point layouts. For quantitative evaluation, we first train a multilayer perceptron network for digit classification, which achieves an accuracy of $98.91\%$ on MNIST test set, and then use the classifier to compute the inception score (Salimans et al., 2016) of 10,000 generated samples. As shown in Table 1, the LayoutGAN with wireframe rendering discriminator achieves a higher inception score than with the relation-based discriminator. To shed light on the relational refinement process of LayoutGAN, we mark each input random point with a location-specific color, and track their refined locations in the generated layouts, as shown in Figure 3b. One can see that colors change gradually along the strokes, showing the network learns some contextually consistent local displacements of the points.

## 4.2 Document layout generation

One document page consists of a number of regions with different element types, such as heading, paragraph, table, figure, caption and list. Each region is represented by a bounding box. Modeling layouts of regions is critical for document analysis, retargeting, and synthesis. In real document data, these bounding boxes are often carefully aligned along the canonical axes, and their placement follows some particular patterns, such as heading always appearing above paragraph or table. Some example layouts are shown in the appendix.

In this experiment, we focus on one-column layouts with no more than 9 bounding boxes that may belong to 6 possible classes as mentioned above. We are interested in how these layout patterns can be captured by our network. For training data, we collect totally around 25,000 layouts from real documents. We also implemented a baseline method that represents the layout by semantic masks as used in Yang et al. (2017); Deka et al. (2017). Specifically, we render all elements into masks and train a DCGAN (Radford et al., 2015) to generate mask layouts in pixels. Then we extract the connected mask regions of each class and output enclosing bounding boxes of all regions. Figure 4 shows some representative generation results (each row consists of 6 samples; high-resolution results can be found in the Appendix). The first row shows the results from DCGAN. DCGAN mixes a layout and its rendering in the generation process. When convolutional layers in DCGAN fails to perfectly replicate the rendering process (which is usually the case), it generates fuzzy and noisy label maps, making the extracted bounding boxes less consistent and less accurate. The third row shows the results from LayoutGAN with wireframe rendering discriminator. We retrieve the most similar real layouts from the training set in the last row as references. It can be seen that LayoutGAN can well capture different document layout patterns with clear definitions of instance-level semantic bounding boxes.

To validate the advantage of wireframe rendering discriminator, we compare the generated results with those from the LayoutGAN with relation-based discriminator. As shown in the second row of Figure 4, the latter can also capture different layout patterns but sometimes suffers from overlapping

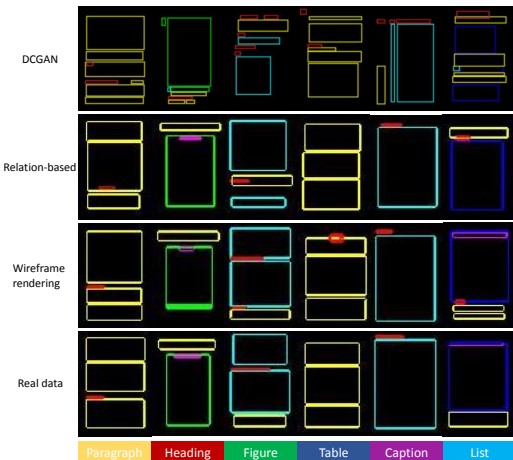

Figure 4: Document layout comparison.

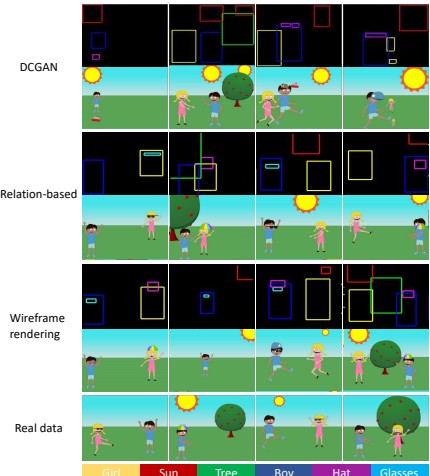

Figure 5: Clipart abstract scene generation.

and misalignment issues. As the bounding boxes in real document layouts are either left- or center-aligned with no overlaps with each other (except for captions with figures or tables), we propose two metrics to quantitatively measure the quality of the generated layouts. The first one is overlap index, which is the percentage of total overlapping area among any two bounding boxes inside the whole page. The second one is alignment index which is calculated by finding the minimum standard deviation of either left or center coordinates of all the bounding boxes. Table 2 provides quantitative comparisons of real layouts and synthesized layouts from LayoutGAN with two different discriminators. One can see that LayoutGAN with wireframe rendering discriminator achieves lower value in both overlapping index and alignment index than the one with relation-based discriminator, validating the superiority of the proposed wireframe rendering method for layout generation. A similar conclusion can also be drawn by analyzing the loss functions. We add shift perturbations to the bounding boxes of real layouts and feed them to both discriminators to examine their loss behaviors. Figure 6 visualizes the loss landscape of both discriminators corresponding to different extent of shift perturbation. One can see that the loss surface w.r.t. shift perturbation of the wireframe rendering discriminator is much smoother than the one of relation-based discriminator.

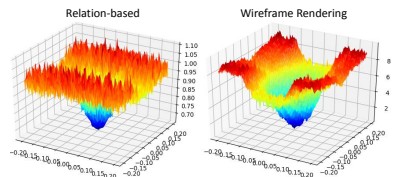

Figure 6: Discriminator loss landscapes.

Table 3: User study results for Clipart abstract scenes.

| Methods | Excellent (%) | Fair (%) | Poor (%) |
|---------|---------------|----------|----------|
| DCGAN | 6.7 | 38.8 | 54.5 |
| Relation | 17.2 | 50.3 | 32.5 |
| Wireframe | 37.3 | 48.0 | 14.7 |

### 4.3 CLIPART ABSTRACT SCENE GENERATION

We now consider generating scenes composed of a set of specific clipart elements. We use the abstract scene dataset (Zitnick et al., 2016) including boy, girl, glasses, hat, sun, and tree elements. To synthesize a reasonable abstract scene, we first use LayoutGAN to generate the layout of inner scene elements by representing each of them as a labeled bounding box with normalized center coordinates, width and height, and flip indicator. Then, we render the corresponding clipart image of each scene element onto a background image according to the predicted position (center coordinates), scale (width and height) and flip, forming a synthesized abstract scene. It is a challenging task as accurate pairwise or higher-order relations of objects are required for synthesizing reasonable scenes, for example, the glasses/hat should be exactly on the eyes/head of a boy/girl with proper scale and orientation. In Figure 5, the first three rows shows the generated layouts and corresponding rendered scenes by DCGAN, LayoutGAN with relation-based discriminator and LayoutGAN with wireframe rendering discriminator respectively. The last row shows some samples rendered from ground-truth scene layouts. It can be seen that, compared to DCGAN and the LayoutGAN with relation-based discriminator, the one with wireframe rendering discriminator can capture pairwise relations of objects precisely, forming more meaningful scene layouts (for example, the glasses/hat accurately lies on the eyes/head of the person with varying scales and flips), validating the superiority of the proposed wireframe rendering strategy. In addition, we conduct a user study involving 20

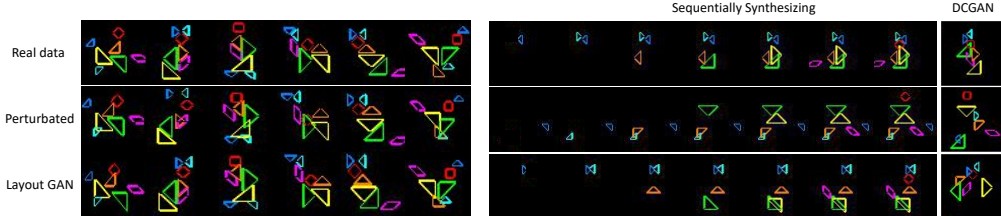

Figure 7: Optimizing perturbed tangram.    Figure 8: Baseline results of tangram design.

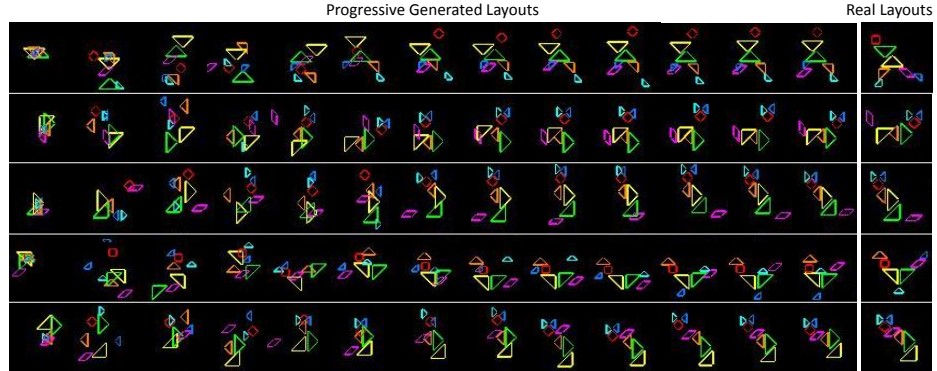

Figure 9: Training progression of tangram graphic design generation (from left to right).

participants for a subjective evaluation of the results. Specifically, we randomly sample 30 abstract scenes synthesized by each of the above three models. A subject is asked to rate the synthetic scenes in terms of three scores (Excellent, Fair, Poor) by following three criteria: 1) whether they have coherent overall structures, 2) whether different objects are placed in reasonable relative positions, scales and flips, and 3) whether there are duplicate objects. We compute the rating percentage of all sampled scenes for each model. As shown in Table 3, the results of LayoutGAN with the wireframe rendering discriminator receive better ratings, which suggests that they are preferred by participants.

### 4.4 TANGRAM GRAPHIC DESIGN

A tangram is a geometric puzzle aiming to form specific shapes using seven pieces of 2D shapes without overlaps. The seven pieces include two large right triangles, one medium right triangle, two small right triangles, one square and one parallelogram, which can be assembled to form a square of side one unit and having area one square unit when choosing a unit of measurement. We collect 149 tangram graphic designs including animals, people and objects. In our experiments, we consider eight rotation/reflection poses for each piece. Given the seven pieces with each initialized by random location and ground-truth pose and class, the LayoutGAN with wireframe rendering discriminator is trained to refine their configurations jointly to form reasonable layouts. Note that it is a challenging task due to the complex configurations and limited data. We perform two experiments. The first one is a perturbation recovery test, in which we first add some random spatial perturbations to seven pieces of real layouts and train a LayoutGAN to recover the original layouts. Figure 7 shows that our network is able to pull back the displaced shapes to the right locations, confirming that the network successfully figures out the relations between different graphic elements. We further train another LayoutGAN to generate tangram designs from purely random initialization. In Figure 9 (high-resolution results can be found in the appendix), each row represents the sampled progressive generated results and the retrieved similar real tangrams. We also present some generated results of DCGAN and a sequential model similar to Wang et al. (2018a) for comparison. One can see from Figure 8 that DCGAN cannot well model the spatial relations among different shapes while the sequential model suffers from accumulated errors. In contrast, the LayoutGAN can generate meaningful tangrams like fox and person, although others may be hard to interpret, as shown in the penultimate row.

## 5 CONCLUSION

In this paper, we have proposed a novel LayoutGAN for generating layouts of relational graphic elements. Different from traditional GANs that generate images in pixel-level, the LayoutGAN can directly output a set of relational graphic elements. A novel differentiable wireframe rendering layer was proposed to rasterize the generated graphic elements to wireframe images, making it feasible to leverage CNNs as discriminator for better layout optimization from visual domain. Future works include adding a content representation, such as text, icon and picture to each graphic element.

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

# 6 APPENDIX

## 6.1 REAL DOCUMENT PAGES

For better understanding the document semantic layouts, we show some layout samples and their corresponding real document pages, as shown in Figure 1.

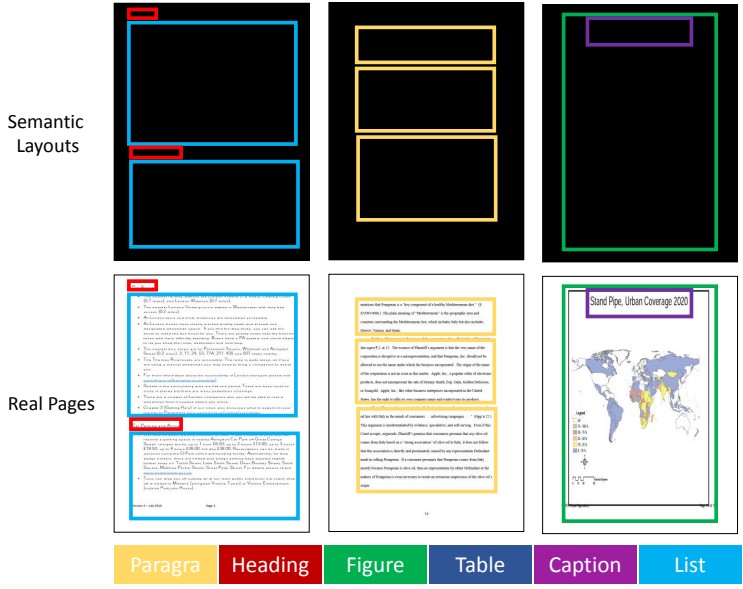

Figure 1: Visualization of document layout samples and their corresponding real document pages.

## 6.2 HIGH-RESOLUTION SYNTHESIZED DOCUMENT LAYOUTS

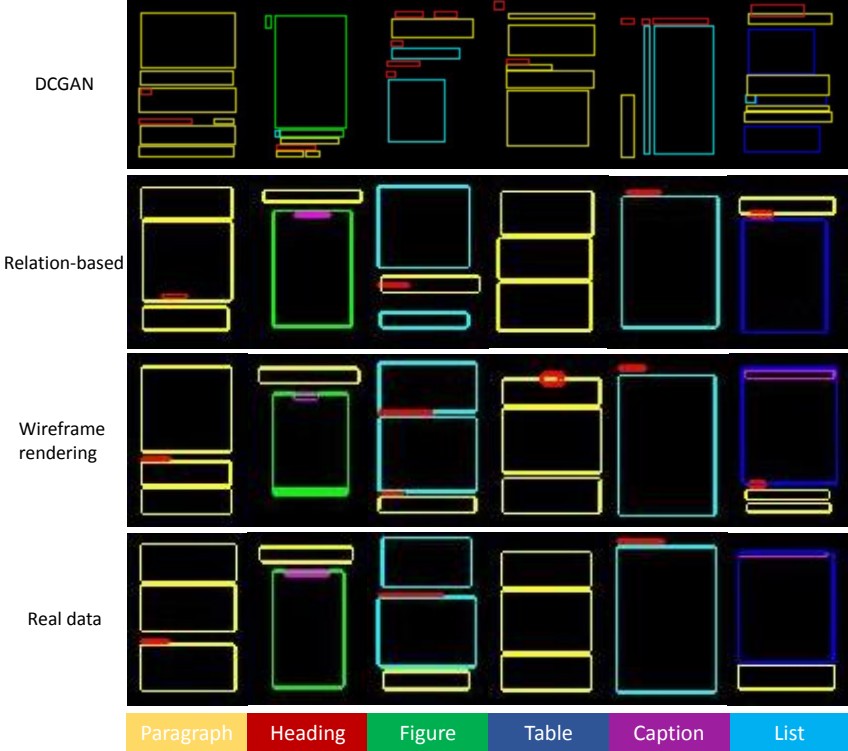

Figure 2: Document layout generation.

## 6.3 HIGH-RESOLUTION SYNTHESIZED CLIPART ABSTRACT SCENES

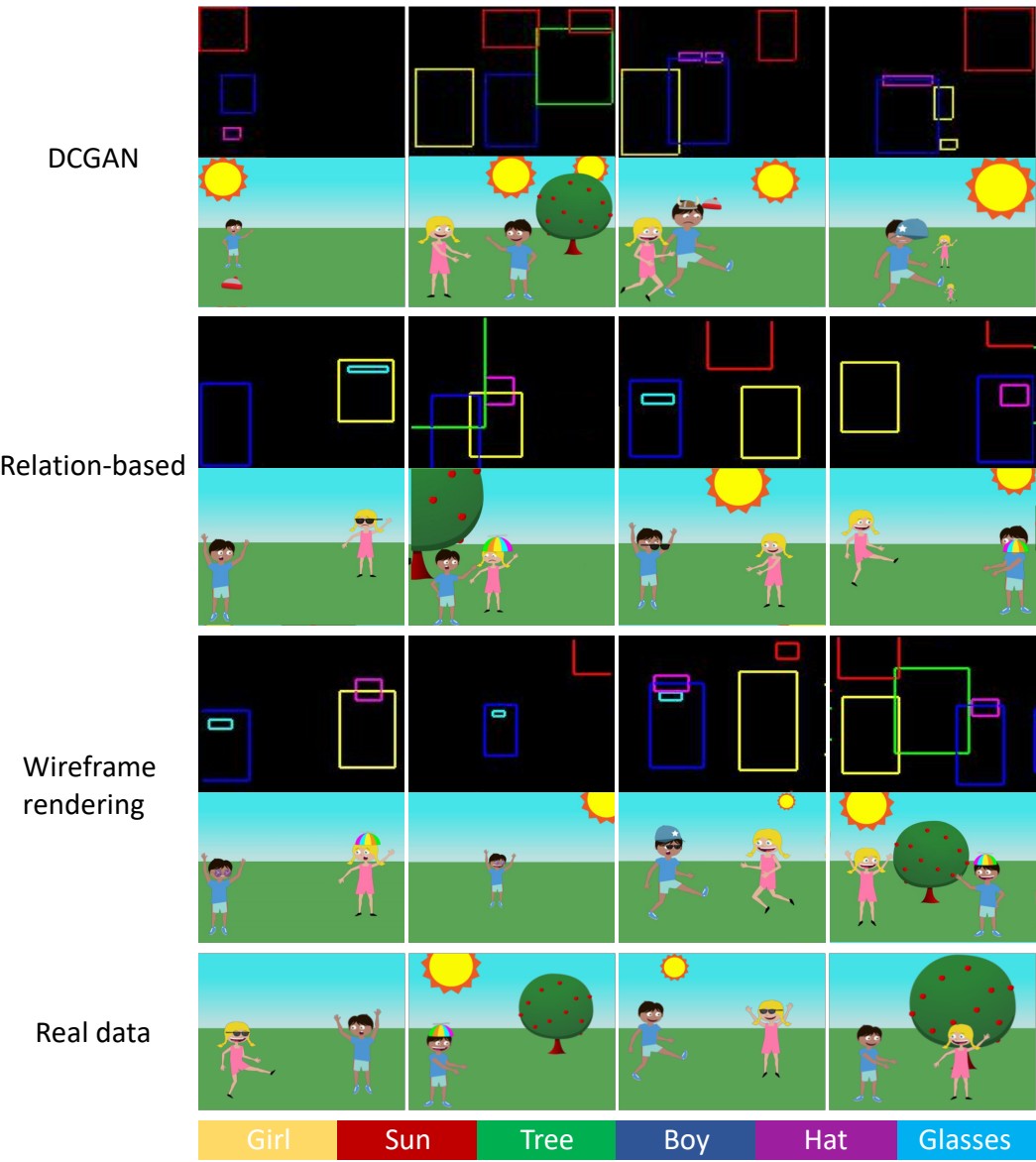

Figure 3: Clipart abstract scene generation.

## 6.4 HIGH-RESOLUTION SYNTHESIZED TANGRAM GRAPHIC DESIGNS

Figure 4: Tangram graphic design.

## 6.5 DETAILED ARCHITECTURE DESIGN OF THE GENERATOR

The generator takes as input a set of graphic elements with random class probabilities and geometric parameters sampled from Uniform and Gaussian distribution respectively. An encoder consisting of three fully connected layers first embeds the semantic and geometric parameters of each graphic element. Two cascaded relation modules implemented as self-attention module followed by a basic residual block ("bottleneck" building block) form one relation block. We cascade two such relation blocks (totally 4 relation modules) for contextual feature refinement. Finally, a decoder consisting of several fully connected layers followed by two heads (implemented as the fully connected layer) with sigmoid activation are used to decode the refined feature of each element back to class probability distributions and geometric parameters respectively.

### 6.6 SYNTHESIZED MOBILE APP LAYOUT DESIGNS

We further conduct a new experiment of mobile app layout design by using the RICO dataset (Deka et al., 2017). For better understanding the mobile app layouts, we first show some layout samples and their corresponding real mobile app screenshots, as shown in Figure 5.

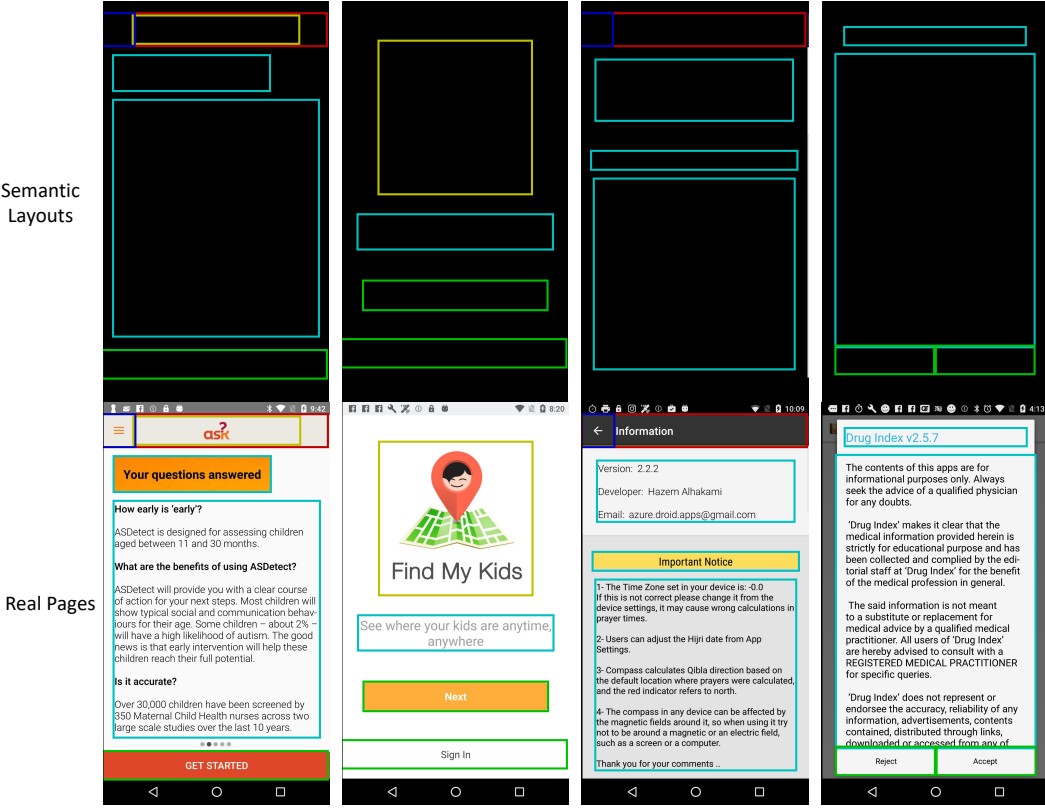

Figure 5: Visualization of mobile app layout samples and their corresponding real mobile app screenshots.

Figure 6 shows the results from LayoutGAN and the retrieved most-similar real layouts from the training set as references. Both the wireframe and the corresponding mask layouts are provided for better visualization.

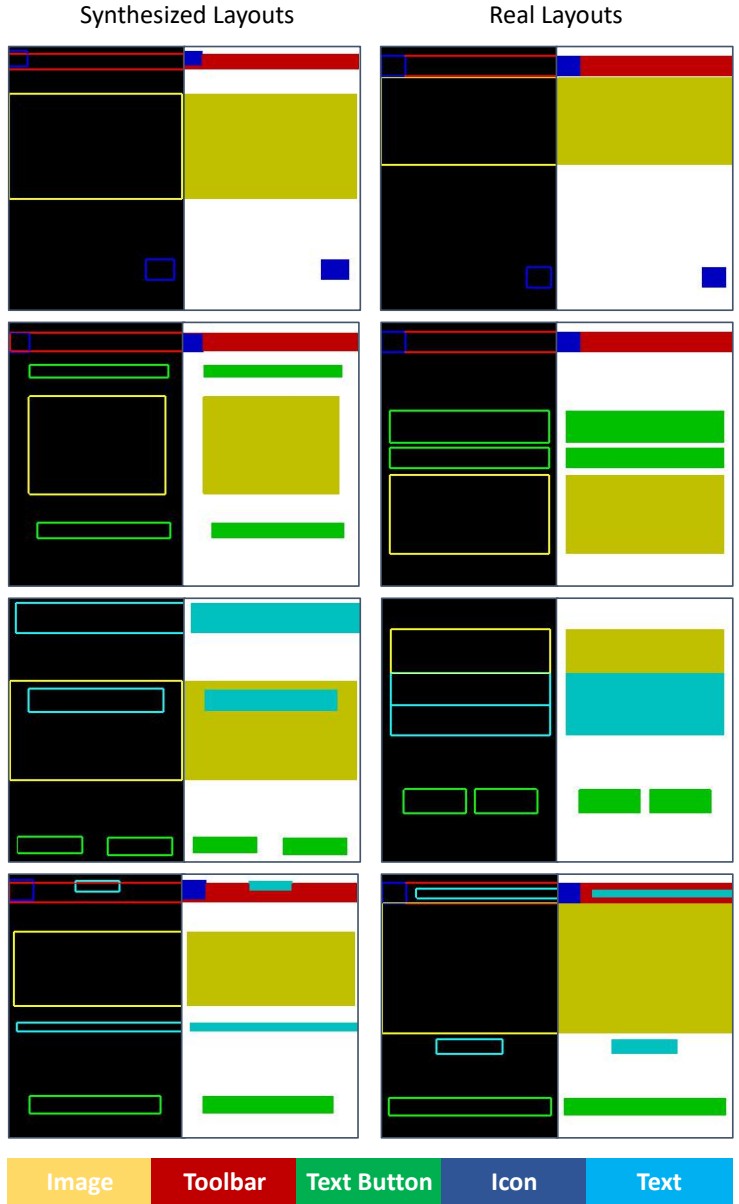

Figure 6: Mobile app layout design.

