# OpenReview forum: "LayoutGAN: Generating Graphic Layouts with Wireframe Discriminators"
_ICLR.cc/2019/Conference_

### Official Review · AnonReviewer3 · 2018-11-03
**Well written paper**

**Rating:** 6
**Confidence:** 4

**Review:**


The authors present a GAN based framework for Graphic Layouts. Instead of considering a graphic layout as a collection of pixels, they treat it as a collection of primitive objects like polygons. The objective is to create an alignment of these objects that mimics some real data distribution.

The novelty is a differentiable wireframe rendering layer allowing the discriminator to judge alignment. They compare this with a relation based discriminator based on the point net architecture by Qi et al. The experimentation is thorough and demonstrates the importance of their model architecture compared to baseline methods.

Overall, this is a well written paper that proposes and solves a novel problem. My only complaint is that the most important use case of their GAN (Document Semantic Layout Generation) is tested on a synthetic dataset. It would have been nice to test it on a real life dataset.

---

> ### Author Response · Authors · 2018-11-22
> **Responses to Reviewer3:**
>
> Q: “My only complaint is that the most important use case of their GAN (Document Semantic Layout Generation) is tested on a synthetic dataset. It would have been nice to test it on a real life dataset.”
>
> A: We added a new experiment of mobile app layout generation by using the RICO dataset (http://interactionmining.org/rico). We showed the results in the appendix due to page limitation. Please see Section 6.6 in the uploaded version for more details.

---

> > ### Comment · AnonReviewer3 · 2018-11-27
> > **Thank you for the new experiments**
> >
> > Thank you for the new experiments. I think this makes the paper stronger.

---

### Official Review · AnonReviewer2 · 2018-11-03
**Interesting application of GANS; Applications on interesting datasets but weak comparison with baselines and quantitative analysis.**

**Rating:** 7
**Confidence:** 3

**Review:**


Summary: This paper presents a novel GAN framework for generating graphic layouts which consists of a set of graphic elements which are geometrically and semantically related. The generator learns a function that maps input layout ( a random set of graphic elements denoted by their classes probabilities and and geometric parameters) and outputs the new contextually refined layout. The paper also explores two choices of discriminators: (1) relation based discriminator which directly extracts the relations among different graphic elements in the parameter space, and (2) wireframe rendering discriminator which maps graphic elements to 2D wireframe images using a differentiable layer followed by a CNN for learning the discriminator. The novel GAN framework is evaluated on several datasets such as MNIST, document layout comparison and clipart abstract scene generation

Pros:
- The paper is trying to solve an interesting problem of layout generation. While a large body of work has focussed on pixel generation, this paper focuses on graphic layouts which can have a wide range of practical applications.
- The paper presents a novel architecture by proposing a generator that outputs a graphic layout consisting of class probabilities and polygon keypoints. They also propose a novel discriminator consisting of a differentiable layer that takes the parameters of the output layout and generates a rasterized image representing the wireframe. This is quite neat as it allows to utilize a CNN for learning a discriminator for real / fake prediction.
- Qualitative results are shown on a wide variety of datasets - from MNIST to clipart scene generation and tangram graphic design generation. I found the clipart scene and tangram graphic design generation experiments quite neat.

Cons:
- While the paper presents a few qualitative results, the paper is missing any form of quantitative or human evaluation on clip-art scene generation or tangram graphic design generation.
- The paper also doesn’t report results on simple baselines for generating graphic layouts. Why not have a simple regression based baseline for predicting polygon parameters? Or compare with the approach mentioned in [1]
- Even for generating MNIST digits, the paper doesn’t report numbers on previous methods used for MNIST digit generation.
Interestingly, only figure 4 shows results from a traditional GAN approach (DCGAN). Why not show the output on other datasets too?

Questions / Remarks:
- Why is the input to the GAN not the desired graphic elements and pose the problem as just predicting the polygon keypoints for those graphic elements. I didn’t quite understand the motivation of choosing a random set of graphic elements and their class probabilities as input.
    - How does this work for the case of clip-art generation for example? The input to the gan is a list of all graphic elements (boy, girl glasses, hat, sun and tree) or a subset of these?
    - It is also not clear what role the class probabilities are playing this formulation.
- In section 3.3.2, it’s mentioned that the target image consist of C channels assuming there are C semantic classes for each element. What do you mean by each graphic element having C semantic classes? Also in the formulation discusses in this section, there is no further mention of C. I wasn’t quite clear what the purpose of C channels is then.
- I found Figure 3b quite interesting - it would have been nice if you expanded on that experiments and the observations you made a little more.

[1] Deep Convolutional Priors for Indoor Scene Synthesis by Wang et al

---

> ### Author Response · Authors · 2018-11-22
> **Responses to Reviewer2-1:**
>
> Q: “While the paper presents a few qualitative results, the paper is missing any form of quantitative or human evaluation on clip-art scene generation or tangram graphic design generation”
>
> A: Thank you for your suggestions. This work is the first attempt to solve layout synthesis from random input for both Clipart scene generation and tangram graphic design (tangram data are collected and annotated by ourselves, we promise to release it upon acceptance). As no previous methods have focused on these problems, there is a lack of widely-accepted quantitative evaluation metrics for both tasks. To this end, we carried out a user study involving 20 respondents for a subjective evaluation of the generated Clipart abstract scenes. Please see Table 3 in the updated version.
>
> Q: “The paper also doesn’t report results on simple baselines for generating graphic layouts. Why not have a simple regression based baseline for predicting polygon parameters? Or compare with the approach mentioned in [1]”
> [1] Deep Convolutional Priors for Indoor Scene Synthesis by Wang et al
>
> A: Thank you for your suggestions. We have supplemented experiments of generating tangram graphic design sequentially as Wang et al [1] for comparison in the updated version. Specifically, Wang et al [1] generate indoor scenes iteratively by adding objects one-by-one. The choice of such sequential paradigm is partly because the rendering process from geometric parameters (object location) to indoor scene images is not differentiable. Similarly, we would have faced such a problem in our layout design. However, we propose a novel wireframe rendering layer to make the layout rendering process differentiable. Benefiting from it, we can predict a set of graphic elements simultaneously in an end-to-end network. But still, we can adopt the sequential paradigm in Wang et al [1] to our layout design problem by generating graphic elements one-by-one. However, we found such sequential synthesis process suffers from accumulated error, which validates the superiority of the proposed LayoutGAN. Please see Figure 8 for comparisons in the updated version.
>
> Q: “Even for generating MNIST digits, the paper doesn’t report numbers on previous methods used for MNIST digit generation.
>
> A: Our experiment on MNIST serves as sanity test. A 2D point, as the simplest geometric form, is not a desirable element representation for our approach, and we do not expect it to compete with other GANs applied to MNIST. We have reflected this in the updated version.
>
> Q: Interestingly, only figure 4 shows results from a traditional GAN approach (DCGAN). Why not show the output on other datasets too?”
>
> A: Thanks for the suggestion. We added experiments to apply DCGAN to both Clipart abstract scene generation and tangram graphic design task in the updated version. Please see Figure 5 and 8.

---

> ### Author Response · Authors · 2018-11-22
> **Responses to Reviewer2-2:**
>
> Q: “Why is the input to the GAN not the desired graphic elements and pose the problem as just predicting the polygon keypoints for those graphic elements. I didn’t quite understand the motivation of choosing a random set of graphic elements and their class probabilities as input.  How does this work for the case of clip-art generation for example? The input to the gan is a list of all graphic elements (boy, girl glasses, hat, sun and tree) or a subset of these?”
>
> A: Given a set of desired graphic elements, our LayoutGAN is actually able to predict their geometric parameters. We demonstrated its ability in the perturbation experiment in Figure 7. However, such synthesis process requires human priors in advance, i.e., the class of each graphic element desired by a reasonable layout.
> Our work goes beyond that. It synthesizes graphic layouts from a set of purely random graphic elements in terms of both geometric parameters and classes. The class of each input element is not predefined but randomly sampled from Uniform distribution, thus the class combination of all the input elements can be in various semantic forms. Take the Clipart experiment as an example, an input set of elements may contain all categories (boy, girl, glasses, hat, sun and tree) or a subset of these with duplicates (boy, girl, glasses, glasses, hat, hat, tree). The model should learn to figure out and adjust the spatial-semantic relations among all elements automatically, and to predict refined class probabilities (can be zero class vector to remove duplicates if necessary) along with geometric parameters for each element to form a reasonable layout.  Predicting class probabilities together with geometric parameters greatly increases the flexibility and applicability of the LayoutGAN on different tasks.
>
> Q: “It is also not clear what role the class probabilities are playing this formulation. In section 3.3.2, it’s mentioned that the target image consist of C channels assuming there are C semantic classes for each element. What do you mean by each graphic element having C semantic classes? Also in the formulation discusses in this section, there is no further mention of C. I wasn’t quite clear what the purpose of C channels is then.”
>
> A: As a graphic element can be boy, girl, hat, glasses, etc, out of C possible classes, we use a C-dimensional vector to represent the probabilities of one element being a particular class. It serves two roles. 1) It allows the network to model the spatial-semantic relations among different elements, e.g. a hat should precisely appear on top of a boy’s head. 2) In the generation experiment, it allows the network to modify the class of each input random element to produce a layout that follow the ground truth class distribution.
> The rendered image consists of C channels because we render wireframes that belong to a specific class with predicted probabilities onto a single channel (C equals to the total number of classes), upon which the CNN-based discriminator can be applied to optimize both the geometric parameters and class probabilities of all graphic elements coherently in a differentiable way. Note that if we render all the wireframes onto a single channel, then we lose the semantic information. In other words, the CNN discriminator would not be able to tell if a bounding box represents a hat or the sun.
>
> Q: “I found Figure 3b quite interesting - it would have been nice if you expanded on that experiments and the observations you made a little more.”
>
> A: Thanks. The colors was used to trace the points from initial random positions to final positions. Following your suggestion, we expanded this experiment on both MNIST and tangram generation by visualizing the displacements or the flows between initial and final positions of graphic elements. In particular, we made animation videos to demonstrate the movements of all the graphic elements. Please review them in the following anonymous link: https://sites.google.com/view/supp-videos-for-iclr-2019/home

---

> > ### Comment · AnonReviewer2 · 2018-11-29
> > **Additional experiments, explanation and comparisons makes it a good paper**
> >
> > Thank you for your detailed rebuttal and for addressing my concerns and responding to my questions.
> > - Specifically I found the additional analysis (both human evaluation and showing results from other baselines) on Clip-art scene generation satisfying.
> > - I also found it helpful to look at DCGAN results for all experiments.
> > - I also looked at the animation videos to demonstrate the movements of all the graphic elements and it helped me appreciate the the approach / design choices a bit more.
> >
> > Overall, I think the paper is greatly improved from the time of the submission and contains more exhaustive evaluation with existing work and shows application for a wide variety of tasks. Based on the rebuttal, I am updating my score to 7 (Good paper)

---

### Official Review · AnonReviewer1 · 2018-11-03
**Interesting application of GAN on Layout rendering**

**Rating:** 7
**Confidence:** 4

**Review:**

Summary:
The paper proposed to use GAN to synthesize graphical layouts. The generator takes a random input and generates class probabilities and geometric parameters based on a self-attention module. The discriminator is based on a differentiable wireframe rendering component (proposed by the paper) to allow back propagation through the rendering module. I found the topic very interesting and the approach seems to make sense.

Quality:
+ The idea is very interesting and novel.

Clarity:
+ The paper is clearly written and is easy to follow.

Originality:
+ I believe the paper is novel. The differentiable wireframe rendering is new and very interesting.

Significance:
+ I believe the paper has value to the community.
- The evaluation of the task seems to be challenging (Inception score may not be appropriate) but since this is probably the first paper to generate layouts, I would not worry too much about the actual accuracy.

Question:
Why not ask the generator to generate the rendering instead of class probabilities?

---

> ### Author Response · Authors · 2018-11-22
> **Responses to Reviewer1:**
>
> Q: “Why not ask the generator to generate the rendering instead of class probabilities?”
>
> A: The generator produces geometric layout parameters together with class probabilities. Rendering a wireframe image from the layout parameters is then trivial. Rendering an application-specific layout, e.g., graphic design bitmap, is application-dependent and unnecessarily complex for modeling layout. Does this answer your question?

---

### Meta-Review · Area_Chair1 · 2018-12-13
**Clear accept ratings from reviewers**

**Confidence:** 4
**Recommendation:** Accept (Poster)

**Metareview:**

Reviewers agree the paper should be accepted.
See reviews below.